# On the Complete Description of Entangled Systems Part II: The (Meta)Physical Status and Semantic Aspects

**DOI:** 10.3390/e24121724

**Published:** 2022-11-25

**Authors:** Karl Svozil

**Affiliations:** Institute for Theoretical Physics, TU Wien, Wiedner Hauptstrasse 8-10/136, 1040 Vienna, Austria; svozil@tuwien.ac.at

**Keywords:** communication cost, simulating quantum correlations, contextuality

## Abstract

We review some semantical aspects of probability bounds from Boole’s “conditions on possible experience” violated by quantum mechanics. We also speculate about emerging space-time categories as an epiphenomenon of quantization and the resulting breakdown of relativity theory by non-unitary and non-linear processes.

## 1. What the Einstein-Podolsky-Rosen (EPR) Conundrum (and This Year’s Nobel Prize for Physics) Is Not about

Many presentations of what is often referred to as “Einstein’s spooky action at a distance” do not grasp the strangeness of the quantum phenomena. You hear statements such as *“Suppose the outcome of Alice’s measurement is random. Suppose further Bob who is far away from Alice (say, spatially separated under strict Einstein locality conditions), also registers a random outcome. Yet, through some kind of `quantum magic’, Bob’s and Alice’s outcomes turn out to be the exact opposite of one another”.* Such claims add to the quantum hocus-pocus that is staged for a lay audience willing to believe that, finally, at the end of a long road of rationality, physics has recovered magic, mysticism, and esoterics.

Indeed, such relational phenomena as described in the previous paragraph, can also be realized by entirely classical means. All that is necessary is that both parties—Alice and Bob—use (hidden) shares that encode opposite directions. This can be conducted even for all conceivable directions Alice and Bob might measure. For instance, Peres introduced the following “bomb” example [1]: consider an (intact) bomb that has an angular momentum of zero initially. Then, the bomb explodes into two fragments with non-zero opposite angular momenta (because of the conservation of angular momentum). These fragments can serve as shares for Alice and Bob. They can measure the dichotomic observables $A_{a}=\mathrm{sgn}\left(J\cdot a\right)$ and $B_{b}=\mathrm{sgn}\left(-J\cdot b\right)$ based on the respective angular momenta J and −J along the directions encoded by (unit) vectors a and b on their side. If Alice’s and Bob’s measurement directions coincide—that is, if a=b—then we obtain a perfect relational encoding: whenever Alices measures the outcome +1, Bob obtains the outcome −1, and vice versa. So if their share—the angular momenta ±J of the bomb fragments—is supposed to be uniformly distributed and occur randomly, then Peres’ exploding bomb can perform ome of the “magic” that is allegedly ascribed to the quanta.

To fully appreciate the quantum strangeness as compared to classical models such as Peres’ bomb scenario, recall that the joint probability of two events *A* and *B* is (symmetrically with respect to exchange of *A* and *B*) defined as the probability P(AandB)=P(AgivenB)P(B)=P(BgivenA)P(A). Furthermore, two events *A* and *B* are (again symmetrically with respect to exchange of *A* and *B*) defined to be independent if the occurrence of *A* has no influence on the probability of *B*, and vice versa; that is, P(BgivenA)=P(B) as well as P(AgivenB)=P(A). Therefore, the aforementioned joint probability of two independent events is the product of their probabilities, that is, P(AandB)=P(A)P(B). Note that this does not exclude relational dependencies on the outcome level: occurrence of *A* “here” could well alter or even completely determine the occurrence of *B* “there”, regardless of the (space-time) distance between *A* and *B*.

Translated into the EPR context, the probability of *A*—say, Alice’s outcome—is in no way altered by any particular outcome *B* on Bob’s side, and vice versa. In the aforementioned bomb example, this is guaranteed by the absence of communication or connection altering the causal [2,3,4,5] relation between its bomb fragments after their (explosion, aka) separation.

If one is willing to invoke relativity theory under strict Einstein locality conditions, then independence can be enforced and certified by a space-like separation of *A* and *B*. Stated pointedly, irrespective of whatever conditions or causes, relative to (the validity of) relativity theory, if *A* and *B* are separated by a space-like interval—such that there is a reference frame in which both events occur at the same time, and there is no reference frame in which the events occur in the same place—their joint probabilities must be the product of their single probabilities.

So far we have implicitly assumed that it makes sense to conceptualize the simultaneous “existence” of value definiteness of *A* and *B* affirmatively. This amounts to the supposition of the “reality” of counterfactual observables (beyond their actual operationalizability) [6]. However, if this cannot be assumed the very notion of their “jointness” or “joint existence” breaks down. One may be tempted to consider such considerations to be sophistry of remote utility. Yet because of the strangeness encountered in the quantum predictions corroborated by the experiments, it might appear prudent to exclude no options for a breakdown of the EPR argument in quantized systems.

Now, for the sake of further pursuing the classical notion of joint probabilities, suppose the simultaneous existence of four pairs of observables pairs (A,B), $\left(A,B^{\prime }\right)$, $\left(A^{\prime },B\right)$, and $\left(A^{\prime },B^{\prime }\right)$ formed by the single observables *A* and $A^{\prime }$ measured by Alice, as well as *B* and $B^{\prime }$ measured by Bob. (There is nothing special about the choice of these joint observables; we might as well have considered only three pairs, or more observable types on Alice’s and Bob’s sides.) Suppose further, to push things to the greatest certification level possible, that all those observations are space-like separated.

Let us again assume that the observables are dichotomic; say they take on the values +1 or −1 (or their respective affine transforms 0 and 1). All conceivable $2^{4}=16$ assignments result in the joint expectations formed by products of pairs AB, $AB^{\prime }$, $A^{\prime }B$, and $A^{\prime }B^{\prime }$ enumerated in Table 1.

If we are lead, by some good intuition or in some systematic ways [7,8,9,10,11], to sum up the first three terms AB, $AB^{\prime }$, $A^{\prime }B$, and subtract the last “double primed” term $A^{\prime }B^{\prime }$, then we arrive at the $AB+AB^{\prime }+A^{\prime }B-A^{\prime }B^{\prime }$ evaluated in the last column of Table 1. Note that, for purely algebraic reasons, all conceivable configurations enumerated result in either +2 or −2, and nothing else. (All possible probabilities and correlations can be obtained by the convex sum of these 16 extreme dispersionless cases.) From this, we conclude that whatever distribution of the $2^{4}=16$ assignments we may consider, the average over all instances of evaluations of $AB+AB^{\prime }+A^{\prime }B-A^{\prime }B^{\prime }$ must fall in-between the $\left[-2,2\right]$ bracket; that is, the average obeys $|\langle AB+AB^{\prime }+A^{\prime }B-A^{\prime }B^{\prime }\rangle |\leq 2$. For historical reasons, this is called the Clauser-Horne-Shimony-Holt (CHSH) inequality [12], in short |〈CHSH〉|≤2, and $CHSH=AB+AB^{\prime }+A^{\prime }B-A^{\prime }B^{\prime }$ is called the CHSH sum.

## 2. What the Einstein-Podolsky-Rosen (EPR) Conundrum (and This Year’s Nobel Prize for Physics) Is about

So what is the EPR conundrum about? To approach an answer, we might concentrate our attention on what happens with the entries of Table 1 if instead of classical shares we use quantum shares. A little min-max calculation [13,14] shows that the quantum average $|\langle CHSH\rangle |=|\langle AB+AB^{\prime }+A^{\prime }B-A^{\prime }B^{\prime }\rangle |\leq 2\sqrt{2}$ exceeds the classical bound 2. How can this happen and manifest itself on an outcome level?

Such a violation of the CHSH inequality (and their variations obtained by permutations of the observables entering the correlation terms) can only be obtained by allowing the correlations and expectations to disobey the product rule from independence; that is, by for instance taking $-A^{\prime }=B=+1$ and $A^{\prime }B=+1$. A quantum simulation of such a situation has been enumerated in the first part [15] of this series of essays. A similar simulation run with 11 trials is presented in Table 2.

As a side note, it might be quite amusing to push these violations of the CHSH inequalities even beyond the quantum [also called Tsirelson (Cirel’son)] bound [16,17,18,19] by non-local exchange of information. These protocols yield violations of independence for every individual trial run, resulting in the algebraically maximal value of the CHSH sum of four.

So, what does a violation of the product rule indicate? One straightforward answer might challenge independence: it might mean that measurement on one side changes the probabilities on the other side. This comes close to Einstein’s motivations for writing the EPR paper [20,21,22]: that the measurement of an outcome on one side “collapses” the state on the other side—it changes the state globally rather than locally—regardless of whether those sides are spatially separated under strict Einstein locality conditions.

Let us recapitulate some (more) possibilities why and how independence, and thus the classical bounds on probability, Boole’s conditions of possible experience, can be violated. (In no way do I claim the following list to be exhaustive.)

(i)There may be non-local communication between Alice and Bob, or between their shares, resulting in some form of contextuality. This can be achieved by either communicating the outcome of Bob’s measurement to Alice (or vice versa) [18,23], or by invoking a non-local machine [24], or by communicating the context in the form of the setting of Bob’s measurement to Alice (or vice versa) [15]. The transactional interpretation [25] offers an alternative approach by considering retarded (forward in time) and advanced (backward in time) flows (of waves or information) from interactions.(ii)Due to complementarity it may be improper to assume the simultaneous co-existence of the observables involved. This is corroborated by (extended) Kochen-Specker type theorems [6,26,27,28,29,30] disallowing any simultaneous co-existence of complementary dispersion-free quantum observables.(iii)Probability theory has to be modified by, for instance, either restricting sets of observables [31], or invoking non-constructive means [32]. We do not discuss these possibilities further than mentioning that physical and formal objections have been raised against them.

My opinion is that there is not the slightest indication that some form of non-local communication happens after the entangled constituents are separated. Moreover, there is not the slightest evidence that the constituents carry hidden variables or hidden shares. So one should take seriously the possibility that those outcomes yielding non-classical correlations are formed (“during” or) through measurements. Nevertheless, the relational encoding of the entangled quantum state enforces a kind of non-locality (through outcome but not parameter dependence) that is “weak” in the sense that it obeys relativity theory—a peaceful coexistence of sorts between quantum and relativity [33,34,35]—in which randomness plays a crucial role. We shall thus sharpen our understanding of this aspect of the EPR conundrum next.

## 3. The Role and Locatedness of Randomness in the Einstein-Podolsky-Rosen (EPR) Conundrum

Let us explore the strangeness of the EPR phenomenology by concentrating on the role randomness plays in it. But first, let us recall that the randomness in classical relational encoding (such as Peres’ bomb example [1]) resides in the uniform stochastic distribution of the parameter(s) of the shares involved.

Quantum mechanics suggests that there are no value definite shares encoded in the constituents of entangled states. Indeed, because of the unitary state evolution (allowing only kinds of, possibly continuous, one-to-one state changes akin to what for a finite number of elements is known as “permutations of order arrangements”), there is a tradeoff—a zero-sum game of sorts—between the relational encoding and the value definiteness of individual constituents. More explicitly, nothing in, say, the four vectors of the Bell basis $|\Phi_{\pm }\rangle =\left(1/\sqrt{2}\right)\left(|++\rangle \pm |--\rangle \right)$ and $|\Psi_{\pm }\rangle =\left(1/\sqrt{2}\right)\left(|+-\rangle \pm |-+\rangle \right)$ indicates that there is more information in these states than the bare (unavoidable) minimum [36]: a 50:50 chance of observing either plus or minus on each one of the two “particles” or shares or fragments. Indeed, these states encode complete value indefiniteness for single share properties; the entire information encoded by the shares is relational. From the point of view of Jaynes’ principle [36]—stating that a “good” encoding is one that “gets rid” of all information not justified or implied by the empirical data—quantum state encoding is “perfect”.

So one might say that, in contradistinction to quantum value indefiniteness, classical randomness is epistemic. The shares are value definite but these definite values are unknown. As the state is chosen (at its preparation) from a continuum of possible states, it contains an infinity of bits of information (with probability one). That is, classical states contain “very much” information. So it might be tempting to state that the classical states appear “over-defined” by requiring the existence of “too many” observables. In contradistinction, the individual properties of a quantum entangled state appears value indefinite. Such a “lean” state may contain no information about the properties of the respective individual quantum constituents whatsoever; that is, it appears “und(erd)efined”.

Let me dwell for one paragraph into the metaphysical status of quantum value indefiniteness [which also applies to (extended) Kochen-Specker theorems]. We might be tempted to ask: if it is not epistemic randomness residing in the value definiteness of the shares, exactly where does the randomness originate? The standard canonical answer might be: as the individual states of the fragments or shares are value indefinite, measurement of such non-existing “properties” must result in irreducible indeterminism [37]. In theological terms, such claim amounts to *“creatio continua”*—the continuous creation of random bits that semantically are without any meaning or message or cause—on a massive scale. In more secular terms, these random bits “come or emerge from nowhere” and are consistent with the assumption of gaps in the physical description [2,3], § III.12-14. [This is the antithesis to the Principle of Sufficient Reason [38], as exposed by Spinoza and Leibnitz, because *“ex nihilo, nihil fit”* (“from nothing, nothing comes”).]

One might try to conceptualize measurements by modeling them by some sort of amplifiers of micro-physical signals (quantum states) to the macroscopic level, as proposed by Glauber’s treatment of cat states [39]. In such a scenario, the contrast or signal-to-noise ratio of the original signal cannot be enhanced; only noise is added.

So if the measurement outcome is entirely due to noise on either side (because there is no signal for the individual states of constituents of the entangled pair), and both sides are (space-like under strict Einstein locality conditions) separated—how come there is a relational encoding? This question is particularly pressing in the case of total alignment of Alice’s and Bob’s instruments on either side.

One might attempt to answer: if one would merely consider the constituents of an entangled pair on the (spatially) separated ends of the EPR setup, one could not understand relational encoding because one would improperly interpret or view the quantum share—the entangled state—from a (separate) observable context. A proper context needs to view the share in its full context, which requires the inclusion of all (in this case two) constituents involved. This is the “message” of inseparable, entangled states of two or more constituent quanta.

## 4. Relativity from Quantum as Epiphenomenon

A caveat is in order: the following section is very speculative and should be perceived accordingly.

In Glauber’s model [39] mentioned earlier, relational encoding amounts to noise that is both correlated with respect to entangled constituents, as well as random with respect to outcomes on either side of an EPR-type setup. Because even as the amplifiers add noise on both sides, they are acting upon a non-local share (such as the vectors of the Bell basis). Therefore, their respective noise is relationally correlated, although the single outcomes appear random. One reason for such a statement is that, within the quantum formalism, there is no a priori means of space-time locatedness; thus entangled pairs do not appear to be separated at all.

From the relational encoding, the “peaceful coexistence” appears straightforward: because for relational encoding, the only information is about the relative states of its constituents, not about the individual constituents. Therefore, no communication of any sort can be performed by variation of the properties of the individual constituents.

This represents a very radical digression from relativity theory; in particular for spatial separability. Some immediate questions arise:(i)When can two outcomes be considered independent and separated? I guess one could attempt to interpret “spatial separation”—two distinct points in space-time—by decomposability of the quantum state; that is, whether the states of the constituents factorize.(ii)Can it occur that, for two (or more) of the same constituents, some of their observables factorize (aka separable, not entangled) and are therefore categorized as “spatially separate or apart”, and other observables are inseparable (aka not factorable, entangled)? This results in a notion of “spatial separateness or apartness” that means relative with respect to the observables continued. Consequently, there is no absolute notion of spatial separation unless all observables are disentangled.(iii)Can it happen that all observables of two quanta are disentangled? My best guess is: for-all-practical-purposes (FAPP [40]) yes, but in principle, no. The situation might be just like for the second law of thermodynamics [41]: if one looks “sufficiently careful”, separability is untenable Because if there has been (in the past and at present) no interaction and, therefore, no entanglement between two outcomes of experiments, there is no connection at all between these events, and they might as well occur in different universes. Consequently, space-time separation appears means relative; and all protocols such as for Poincaré-Einstein synchronization [42,43] are means relative.(iv)Is it possible to generate emerging space-time categories such as frames or coordinatizations, by purely quantum mechanical means?

## 5. Breakdown of Relativity Theory from Non-Unitary and Non-Linear Processes

Again, a caveat is in order: the following section is very speculative and conjectural.

Because of the linear, unitary quantum evolution —essentially restricting quantum evolution to one-to-one processes—the no-cloning theorem (beyond a single context [44], pp. 39–40) blocks instantaneous or space-like (aka faster-than-light) communication [39,45,46,47,48] by disallowing the replication of quanta in states associated with complementary observables.

And yet, might it be possible to perform such tasks in, say, a non-unitary or in non-linear frameworks, such as in non-linear optics? The aforementioned no-cloning theorem would not apply to such capacities, if they exist.

For the sake of examples, I have put forward a scheme based on the stimulated emission of photons in the presence (or absence) of “a large number” of photons in identical states [49]. The Bose statistics demands stimulated emissions to occur more likely in the presence of other identical photons. More precisely, the probability that a photon will be absorbed in the presence of *n* photons (in identical mode) is ${n|a|}^{2}$, where *a* stands for the probability amplitude of absorption of a single photon when no other photon (in identical mode) is present [50], § 4.4. This photon bunching corresponds to, say, electron or neutron anti-bunching: A similar “inverse” argument holds for quanta obeying Fermi statistics.

One might argue against such scenarios by stating that such statistical processes can occur only at the site of the production of particles; in this case, for instance, inside the non-linear crystal emitting two photons in a singlet state. However, delayed-choice experiments and the aforementioned non-local aspects of entanglement might put such counter-arguments in question. Of course one has to be cautious and remember that, “when analyzing *…* we can fall into all kinds of traps” by rectifying—“to take too realistically—concepts such as wave and particle” [51].

## 6. Summary

We have reviewed several (mis)conceptions regarding the EPR framework. One point made is that relationality, that is, (perfect) correlations among the constituents of once interacting particles, or partitions of objects that were once inseparable, is neither mindboggling nor “spooky”. Rather the quantum “advantage” or distinction over classical systems lies in the particular modulation of the respective correlations. This comes about by the particular probabilities of quantum mechanics, which need to be based on Hilbert space entities such as vectors “viewed from” orthonormal basis frames or contexts [52]. The quantum shares are state-as-vectors, and when confronted with the respective orthogonal projection operators corresponding to elementary yes-no propositions [53], they are mapped into the well-known trigonometric (e.g., cosine) correlations. If (unless perpendicular) there is a “mismatch” between state vectors and the vector corresponding to the projectors (formed by the dyadic product of the respective vectors), this amounts to “stronger-than-classical” correlations, which in turn result in violations of Boole’s conditions of possible experience for probabilities based upon observables representable by a single Boolean algebra or by partition logic (a set representable pasting of Boolean algebras).

This view also entails a modification of the space-time concept—in particular, the adjustment of an a priori Kantian framework of space-time—very much in the spirit of Einstein’s operationalization of synchronicity and space-time frames [42]. What is usually considered as “non-locality” is re-interpreted as an artifact of the intrinsic perception of a “vector world”, when confused with the inappropriate idealization of a “classical” universe conceptualized by (power) sets, and set-theoretical operations, such as the formation of set-theoretical intersections and unions.

We have also speculated that perhaps not the last word is spoken about the current no-cloning concepts. While these are unavoidable consequences of, and undoubtedly valid relative to, their premises and the means involved, in particular, linearity and one-to-one unitary state evolution that essentially amounts to a “continuous permutation” of the quantum state, any digression or allowance of, say, non-linear means, could give rise to an end of Shimony’s “peaceful co-existence” [33,35] of quantum mechanics and relativity theory. In such a case, a rather straightforward choice would be to conceptualize space-time categories by grounding them in quantum mechanical means and terms.

## Figures and Tables

**Table 1 entropy-24-01724-t001:** Peres-type valuation table enumerating all $2^{4}=16$ potential outcomes of four events in the second to fifth columns, joint outcomes (assuming independence) and thus products of certain pairs of outcomes in the sixth to ninth colums, and the resulting Clauser-Horne-Shimony-Holt summations $CHSH=AB+AB^{\prime }+A^{\prime }B-A^{\prime }B^{\prime }$ in the last column.

Valuation #	*A*	$A^{\prime }$	*B*	$B^{\prime }$	AB	${\mathrm{AB}}^{\prime }$	$A^{\prime }B$	$A^{\prime }B^{\prime }$	CHSH
1	+1	+1	+1	+1	+1	+1	+1	+1	2
2	+1	+1	+1	−1	+1	−1	+1	−1	2
3	+1	+1	−1	+1	−1	+1	−1	+1	−2
4	+1	+1	−1	−1	−1	−1	−1	−1	−2
5	+1	−1	+1	+1	+1	+1	−1	−1	2
6	+1	−1	+1	−1	+1	−1	−1	+1	−2
7	+1	−1	−1	+1	−1	+1	+1	−1	2
8	+1	−1	−1	−1	−1	−1	+1	+1	−2
9	−1	+1	+1	+1	−1	−1	+1	+1	−2
10	−1	+1	+1	−1	−1	+1	+1	−1	2
11	−1	+1	−1	+1	+1	−1	−1	+1	−2
12	−1	+1	−1	−1	+1	+1	−1	−1	2
13	−1	−1	+1	+1	−1	−1	−1	−1	−2
14	−1	−1	+1	−1	−1	+1	−1	+1	−2
15	−1	−1	−1	+1	+1	−1	+1	−1	2
16	−1	−1	−1	−1	+1	+1	+1	+1	2

**Table 2 entropy-24-01724-t002:** Peres-type valuation table of a quantum simulation (by non-local context communication) whose CHSH sum converges, in the limit of many trials, towards the maximal quantum violation $2\sqrt{2}$ of the CHSH inequality.

Trial #	*A*	$A^{\prime }$	*B*	$B^{\prime }$	AB	${\mathrm{AB}}^{\prime }$	$A^{\prime }B$	$A^{\prime }B^{\prime }$	CHSH
1	+1	−1	+1	+1	+1	+1	+1	−1	4
2	+1	+1	+1	+1	+1	+1	+1	−1	4
3	+1	+1	+1	+1	+1	+1	+1	−1	4
4	+1	+1	+1	+1	+1	+1	+1	−1	4
5	+1	+1	+1	+1	+1	+1	+1	+1	2
6	+1	−1	+1	+1	+1	+1	+1	−1	4
7	+1	+1	+1	+1	+1	+1	+1	−1	4
8	+1	−1	−1	−1	−1	−1	−1	−1	−2
9	+1	−1	−1	−1	−1	−1	−1	−1	−2
10	+1	−1	−1	−1	−1	−1	−1	−1	−2
11	+1	−1	+1	+1	+1	+1	+1	−1	4
⋮	⋮	⋮	⋮	⋮	⋮	⋮	⋮	⋮	⋮

## Data Availability

Not applicable.

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
