# Peer review of "On the Complete Description of Entangled Systems Part II: The (Meta)Physical Status and Semantic Aspects"

_entropy, 2022, doi:10.3390/e24121724_

Round 1

Reviewer 1 Report

This short paper is one among the many works that are constantly appears in the pertinent scientific journals concerning the ontological nature of quantum mechanics and its semantic interpretation. Just like most of these works, more than giving answers raises questions!

However, I believe that the paper is reasonably well written and with its own logic. Therefore, I have no objection to its publication in its current form.

Author Response

I agree with the reviewer that this is a paper (among many because of the challenge) trying to address the "meaning" of the subject.

It contributes to the debate with the particular means involved. 

Reviewer 2 Report

The manuscript is a metaphysical overview of the phenomena associated with entanglement. As a result of the review, some possible ideas in the interpretation of quantum information from the point of view of relativity theory and the nature of space-time are suggested. The author confesses very speculative character of  these suggestions.

I do not see any there is no iformal mistakes in the text,  except for a slight inaccuracy in lines 220-221: " there is no interaction and, therefore, no entanglement between two outcomes of experiments". In fact, there may not be any interaction now, as long as it was in the past.  It is easy to correctand and I suppose that manuscript  can be accepted almost as it is.

However I strongly recommend to the author to iclude in the text and to express his attitude to the following works:

1. Transactional interpretation of entanglement by J. Cramer.  The possibility of quantum information propagation both in forward and reversed time exclude questions about "instnenious" correlations and contradiction with relativity theory.

2. Entanglement does not exclude causality namely: asymmetric mixed states are causal, and quantum causality has richer and more interesting properties than classical. This is the subject of a series of articles by E. Kiktenko 2010-2015.

3. The idea that space-time is not primary,  but can be constructed on the basis of binary relations is not new.  In particular, it is developed in the works of Yu. Vladimirov

Author Response

I would kindly like to thank the Reviewer for pointing out a lapsus that I herewith correct by stating

"Because if there has been (in the past and at present) no interaction and, therefore, no entanglement between two outcomes of experiments ..."

In accordance with the Referee's suggestions I have added the following references:

1. "The transactional interpretation~\cite{Cramer-RevModPhys.58.647} offers an alternative approach by considering retarded (forward in time) and advanced (backward in time) flows (of waves or information) from interactions."

2. a paper on causality in quantum mechanics by Korotaev, Kiktenko, Amoroso, Rowlands, and Jeffers. I have also added a book on this topic by  Judea Pearl.

Reviewer 3 Report

This paper is a philosophical discussion of what the violation of the CHSH form of the Bell inequality means, and the ways we should (and should not) think about it. It is a subject of great interest, particularly in a year when the first Nobel prize has been awarded in the subject to three researchers who have made major contributors to it. Svozil himself has contributed to the theoretical discussion of the inequalities over the years, and this is the second in a series of papers by him to draw some lessons from work on it. I find his views interesting, and while questionable and debatable in places, also well put and thought provoking. I think the paper is likely to be read with interest by a number of readers and so I would recommend publishing it.

Author Response

Thank you for this kind evaluation.